# Genome-Wide Association Study for Milk Somatic Cell Score in Holstein Friesian Cows in Slovenia

**DOI:** 10.3390/ani14182713

**Published:** 2024-09-19

**Authors:** Ali Ashja, Minja Zorc, Peter Dovc

**Affiliations:** Biotechnical Faculty, University of Ljubljana, 1000 Ljubljana, Slovenia; a.aliashja@gmail.com (A.A.); minja.zorc@bf.uni-lj.si (M.Z.)

**Keywords:** mastitis, somatic cell count, GWAS, SNP array

## Abstract

**Simple Summary:**

Mastitis is a major issue in dairy farming, leading to significant economic losses and poor milk quality. This study aimed to find genetic markers associated with resistance to mastitis by analyzing the somatic cell score (SCS) in Slovenian Holstein Friesian cows. A genome-wide association study (GWAS) was conducted on genotypic and phenotypic data of dairy cows from a single farm in Slovenia. The study identified five significant genetic markers associated with somatic cell score. These markers were close to six already known candidate genes. The findings suggest that these genetic markers could be used in breeding programs to improve mastitis resistance in dairy cows, potentially leading to healthier herds and better milk production.

**Abstract:**

Mastitis is a serious challenge for the dairy industry, leading to economic losses and affecting milk quality. The aim of this study is to identify genetic factors associated with mastitis resistance by conducting a genome-wide association study (GWAS) for the somatic cell score (SCS). Phenotypic records of 350 Holstein Friesian cows were obtained from the Slovenian Cattle Recording Scheme Database and consisted of around 1500 lactation data from 2012 to 2023 collected on a single farm in Slovenia. Corresponding genotypic data were also retrieved from the same database and genotyped using the Illumina BovineSNP50 BeadChip (Illumina, Inc., San Diego, CA, USA). For the association study, three SCS parameters were considered, including lactation mean somatic cell score (LM_SCS), maximum SCS value (SCSMAX), and top three mean value of SCS (TOP3). After performing a GWAS using FarmCPU and BLINK models, five significant SNPs associated with the TOP3 trait were found on BTA 14, 15, 22, and 29. The identified SNP markers were closely linked to six known candidate genes (*DNASE1L3*, *SLC36A4*, *ARMC1*, *PDE7A*, *MMP13*, CD44). These results indicate potential genetic markers associated with SCS in the Slovenian Holstein Friesian population.

## 1. Introduction

Mastitis is a mammary tissue inflammation that leads to significant economic losses in the contemporary dairy industry and affects the quality of dairy products globally [1]. Intensive farming practices for high milk yield and a positive genetic correlation between mastitis susceptibility and milk yield have led to higher occurrence of mastitis in modern dairy production [2,3]. Breeding can help improve resistance to mastitis, but the low heritability of the trait and difficulties in assessing it directly, based predominantly on the binary phenotypic data, pose significant challenges [4]. Therefore, improving the understatement of the genetic base of mastitis and its indicator traits represents a significant goal for the dairy cattle breeding industry. The somatic cell count (SCC) is considered the most suitable indicator trait for mastitis resistance due to its high heritability, medium-to-high genetic correlation with mastitis, and ease of measure [5,6,7]. In many countries, SCC is log-transformed to somatic cell score (SCS) to achieve the normal distribution and is evaluated as the only trait contributing to udder health [8,9]. However, with a moderate genetic correlation between SCC and mastitis, several studies have suggested to use alternative traits derived from SCC, which have a higher correlation with mastitis, to capture different aspects of mastitis and to improve genetic evaluation for mastitis resistance. Some novel alternative traits investigated by these studies include lactation average SCS over early (SCS150) [10,11,12] and late (SCS151–305) lactation [10,11,12], maximum SCS (SCSMAX) [10,13], and standard deviation of SCS (SDSCS) [10,14]. In addition, excessive test-day (TD) SCC (TDSCC) and pattern of peak in SCC (PK) were used by de Haas et al. [15], Koeck et al. [10], and Urioste et al. [11]. TDSCC describes the dynamics of SCC based on the defined threshold, such as TD > 200 (presence of at least one test-day SCC above 200,000 cells/mL) and TD > 500. Nani et al. [13] used the top three values of SCS for a single animal (TOP3) for the first time to capture repetitive high values of SCS. The lactation-average somatic cell score has traditionally been utilized to enhance genetic resistance to mastitis [15]. However, it has been shown that alternative SCC traits offer valuable insights for the genetic evaluation of mastitis, which cannot be captured using mean SCC. Genetic correlations between mastitis and various SCC traits, including SCSMAX, standard deviation of SCS (SDSCS), TD > 200, TD > 500, and PK were found to be surprisingly high, ranging from 0.82 to 0.91 [10]. Green et al. [14] revealed that when considering a range of pathogens, the SCSMAX and SDSCS of log SCC were more effective clinical mastitis (CM) indicators than the mean log SCC.

Mastitis is a complex and multifactorial disease influenced by the intricate interaction between genetic, physiological, and environmental factors, including pathogenic bacteria. Therefore, improving mastitis resistance is difficult as it is challenging to identify mastitis-associated alleles in livestock [16,17,18]. The advent of high-throughput genotyping technologies has enabled the execution of genome-wide association studies (GWAS) in livestock, facilitating the identification of significant SNPs associated with complex traits and leading to the discovery of quantitative trait loci (QTL) and candidate genes, which are essential in enhancing economically important traits across different breeds and populations [19,20]. In dairy cattle, several studies have utilized GWAS to investigate the association of SNPs with clinical mastitis (CM) and SCC as its indicator trait to provide new insights into the genetic architecture of resistance to mastitis [4,13,20,21]. For instance, Nani et al. [13] genotyped 544 Argentinian dairy cattle using Illumina BovineSNP50v2 BeadChip and identified six SNPs associated with traits derived from SCC, highly correlated with mastitis. In addition, the study identified candidate genes around significant SNPs with potential roles in immune response. In another study, Ilie et al. [20] conducted a GWAS for SCS on 723 Romanian spotted (RS) and Romanian brown (RB) cattle using Axiom Bovine v3 SNP-chip and found four candidate genes (*HERC3*, *LUZP2*, *AKAP8*, and *MEGF10*) associated with SCS, which were previously reported in other studies. Zhou et al. [22] conducted GWAS to analyze the genetic architecture of mastitis resistance using SCS as an indicator trait for udder health. They identified three significant SCS-associated SNPs on chromosomes BTA 5, 22, and 8 near candidate genes, including *DYRK2*, linked to udder support. Meredith et al. [23] genotyped 702 Holstein Friesian bulls, which are widely used for artificial insemination in Ireland, to identify genetic loci associated with SCS. The study identified 28 QTL regions and several candidate genes. The most significant QTL on chromosome 6 contained a small non-coding RNA gene linked to autoimmune response. Other identified regions included a cluster of β-defensin genes on chromosome 13, which is important for innate immunity.

Based on data from the entire Slovenian population of Holstein Friesian cows, the introduction of a new somatic cell count index (SCCI) was proposed for the accurate prediction of milk yield losses caused by elevated SCC [24]. However, despite the widespread use of GWAS to identify genetic variants linked to mastitis and its indicator traits, such as SCS, there are, to the best of our knowledge, no studies that have explored the genetic architecture of SCS in Slovenian Holstein Friesian cows. Most of the previous GWAS studies in this population have focused on other traits; for example, Yin et al. [25] performed a GWAS on behavioral traits in local dual-purpose cows, including Slovenian Holstein and Slovenian Brown Swiss cattle.

The present study aims to further investigate the genetic architecture underlying mastitis resistance in dairy cattle by utilizing a GWAS. Building upon previous research, we analyzed the association of single-nucleotide polymorphisms (SNPs) with SCS and other SCC-derived traits in Slovenian Holstein Friesian cows. We aimed to identify new candidate genes associated with SCS and mastitis resistance utilizing high-throughput genotyping technology and comprehensive phenotypic data. We hypothesize that specific SNPs are significantly associated with the somatic cell score and its alternative traits (LM_SCS, SCSMAX, and TOP3), reflecting different aspects of mastitis susceptibility and resistance. Furthermore, we predict that candidate genes located in the vicinity of significant SNPs are involved in immune response mechanisms that influence the development of mastitis. Our results contribute to developing more effective breeding strategies for enhancing udder health and improving the economic sustainability of the dairy industry.

## 2. Materials and Methods

### 2.1. Phenotypes

The study was conducted on 350 Holstein Friesian (HF) cows from a single farm in Slovenia. All cows were kept under similar sanitation, management, and housing conditions, ensuring a uniform environment for all animals included in the study. Phenotypic data consisting of 1500 daily milking records from 2012 to 2023 were obtained from the Slovenian Cattle Recording Scheme Database. Only SCC records of the first two lactations were considered for this study.

Initially, our phenotypic dataset consisted of 13,735 test-day records. After filtering the data to include only the first two lactations, 6864 test-day milk records were used for this study, with 3497 records (approximately 51%) from the first lactation and 3367 records (approximately 49%) from the second lactation.

#### Definition of Alternative SCC Traits

To achieve a normal distribution of SCC records, we applied a logarithmic transformation to the daily SCC values through the equation SCS = log_2_ (SCC/100) + 3 to transform SCC to SCS [8]. Based on various studies, three alternative SCS traits were defined to capture different aspects of mastitis. The first trait, lactation mean SCS (LM_SCS), was calculated by averaging the test-day SCS records for each cow in the first and second lactation. The second trait, maximum SCS (SCSMAX), was determined by identifying the highest SCS value recorded for each cow, highlighting peak levels of somatic cells indicative of severe mastitis episodes [10]. The third trait, top 3 mean SCS (TOP3), was the arithmetic mean of each cow’s top 3 highest SCS values [13]. These alternative traits were designed to provide a comprehensive understanding of mastitis from multiple perspectives, aiding in the identification of genetic factors associated with different aspects of mastitis susceptibility and udder health in dairy cows. These traits offer a more nuanced understanding of somatic cell count (SCC) dynamics than simply using the lactation average [11,12,15].

### 2.2. Genotyping and Quality Control 

Genotypic data were collected from the Slovenian Cattle Recording Scheme Database, obtained by genotyping using the Illumina Bovine SNP50 Beadchip (Illumina, Inc., San Diego, CA, USA). Quality control on genotyped samples was performed using PLINK v1.9 [25]. SNPs with a minor allele frequency (MAF) of less than 0.05 and those with a call rate of less than 0.90 were excluded. Additionally, individuals with more than 10% missing genotype data were removed. SNPs that did not conform to Hardy–Weinberg equilibrium (*p* > 106) were also eliminated. After these quality control measures, 39,907 out of the initial 43,044 SNPs and 343 out of 350 cows were included in the subsequent analysis.

### 2.3. Linkage Disequilibrium

Linkage disequilibrium (LD) was quantified using the PLINK software v1.9 [26]. The LD measure *r*^2^ was computed for all SNP pairs on each chromosome following the method described by Hill and Robertson [27]. The formula for *r*^2^ is given by
$$r^{2}=\frac{D^{2}}{f\left(A\right)\times f\left(a\right)\times f\left(B\right)\times f(b)}$$
where $D=f\left(AB\right)-f\left(A\right)f\left(B\right)$. Here, f(AB) represents the observed frequency of the haplotype AB, and f(A), f(a), f(B), and f(b) denote the observed frequencies of alleles *A*, *a*, *B*, and *b*, respectively.

### 2.4. Genome-Wide Association Analysis 

For GWAS of SCS traits, we used the GAPIT R package version 3.0. Two GWAS methods were employed: fixed and random model circulating probability unification (FarmCPU) and Bayesian information and linkage disequilibrium iteratively nested keyway (BLINK). The FarmCPU method addresses the issue of confounding factors by combining fixed- and random-effects models. It iteratively adjusts for population structure and kinship, thereby improving the power to detect true associations while controlling for false positives [28]. Population structure was accounted for in the model through the use of principal components (PCs) derived from principal component analysis (PCA). These PCs were included as covariates in the GWAS model to correct for population stratification. BLINK is the improved version of the FarmCPU model in which Bayesian information criteria (BIC), which iteratively select SNPs likely to be associated with the trait, and linkage disequilibrium approaches are used [29].

The significance threshold for the GWAS was set at a −log_10_ (*p*) value of 6.0, which corresponds to a *p*-value of 1 × 10^−6^. This threshold was selected based on the Bonferroni correction for multiple testing, which is calculated by GAPIT3 to control for false positives while maintaining the power to detect significant associations. The SCS traits analyzed included LM_SCS, SCSMAX, and TOP3 during the first and second lactation. Manhattan plots were generated using the internal functions within GAPIT. These methods allowed for the efficient detection of genetic variants associated with SCS traits, utilizing mixed linear model (MLM) capabilities to account for population structure and kinship, thereby reducing false positives and providing insights into the genetic basis of mastitis in cattle.

### 2.5. Candidiate Gene Identification

We used SNPchiMp v.3 tools [30] to identify the genomic positions of significant markers in the current bovine genome assembly (ARS-UCD1.3). Subsequently, candidate genes were identified using the Bioconductor package biomaRt in R [31] to query the Ensembl database (https://www.ensembl.org/index.html, accessed on 30 June 2024) for genes within 1 Mb upstream and downstream of significant SNPs [32] based on the Bos taurus genome assembly (ARS-UCD1.3). We also used the human gene database GeneCards (https://www.genecards.org, accessed on 2 July 2024) to perform functional annotations of identified genes.

## 3. Results

### 3.1. SNP Distribution across Chromosomes

This study comprehensively examined the genotypic characteristics of SNP markers across Bos taurus chromosomes, as detailed in Table 1. The largest chromosome, BTA1, spanned 158.53 Mb and hosted the highest number of SNPs at 2532, offering an SNP density of 15.97 SNPs/Mb. In contrast, the smallest chromosome, BTA25, which is only 42.35 Mb in length, contained 769 SNPs, resulting in the highest SNP density observed at 18.16 SNPs/Mb. A total of 83 SNP markers were excluded from the subsequent analysis due to their undefined position within the ARS-UCD1.3 assembly. Notably, the SNP density varied widely, with the lowest recorded at 2.32 SNPs/Mb on chromosome X and the highest at 18.3 SNPs/Mb on BTA19. The mean distance between SNPs and their standard deviation was also calculated for each chromosome, providing a measure of genomic clustering. For instance, BTA19, while having the highest SNP density, had a mean SNP distance of 54.78 Kb with a standard deviation of 54.44 Kb. Furthermore, chromosome X, which exhibited the lowest SNP density, showed an exceptionally high mean SNP distance of 459.62 Kb and a substantial standard deviation of 498.24 Kb.

### 3.2. Descriptive Statistics of SCS Traits

The descriptive statistics of the SCS traits are presented in Table 2. The results indicate that both, the mean and variability of SCS increased in the second lactation. For LM_SCS in the first lactation (LM_SCS_1_), the mean SCS was 2.49, with a standard deviation (SD) of 1.07, indicating moderate variability. The coefficient of variation (CV) was 0.43, with SCS values ranging from 0.48 to 5.80. In the second lactation (LM_SCS_2_), the mean SCS increased to 2.87, the SD rose to 1.31, and the CV was at 0.53, with SCS values spanning from 0.10 to 7.63. This comparison highlights that somatic cell counts tend to increase and become more variable as cows progress from their first to the second lactation.

Similarly, the SCSMAX traits showed increased mean and variability in the second lactation. For the SCSMAX in the first lactation (SCSMAX_1_), the mean was 4.51 with an SD of 1.67 and a CV of 0.37, with values ranging from 1.40 to 9.97. In the second lactation (SCSMAX_2_), the mean maximum SCS increased to 4.99, the SD rose to 1.90, and the CV was 0.42, with values ranging from 0.53 to 9.90. This indicates an increase in the average maximum somatic cell count and variability during the second lactation.

Analyzing the TOP3 traits, the mean SCS during the first lactation (TOP3_1_) was 3.72, with an SD of 1.28 and a CV of 0.34 and values ranging from 1.53 to 8.75. In the second lactation (TOP3_2_), the mean SCS increased to 4.10, with an SD of 1.42 and a CV of 0.35 and values ranging from 1.33 to 9.53. This analysis indicates that the highest SCS readings tend to be higher and more variable during the second lactation.

### 3.3. Population Structure Analysis

Population structure was examined using principal component analysis (PCA), and the first three principal components (PCs) were included as covariates in the GWAS model to adjust for population stratification. The PCA plot (Figure 1) shows the distribution of individuals based on the first two principal components, indicating the presence of population structure among the analyzed Slovenian Holstein Friesian cows.

### 3.4. Association of SNPs with SCS Traits

In this study, a GWAS was conducted on three traits derived from SCS using the BLINK and FarmCPU models. We identified several single-nucleotide polymorphisms significantly associated with the TOP3 trait in the first lactation (TOP3_1_), considering SNPs with *p*-values less than 10^−6^ as significant. The BLINK model detected five significant markers associated with the TOP3 trait in the first lactation on BTA14, BTA15, BTA22, and BTA29. Similarly, the FarmCPU model identified two significant markers for the TOP3_1_ trait on BTA22 and BTA29. Remarkably, the markers on BTA22 and BTA29 were located at the same positions and had identical rs numbers in both models. The Manhattan plots for the GWAS of SCS traits with FarmCPU and BLINK models are presented in Figure 2.

### 3.5. Linkage Disequilibrium

LD decay was analyzed to determine the appropriate genomic distance for identifying candidate genes in our population. The decay pattern was evaluated using the R^2^ values between SNP pairs at various genomic distances. In our dataset, the LD decay pattern was assessed with a focus on the R^2^ thresholds of 0.2 and 0.3. The analysis revealed that the mean R^2^ values did not drop below 0.2 within the examined distance range, indicating a relatively high level of LD across these distances. However, a significant drop in R^2^ values was observed at a threshold of 0.3. Specifically, the mean R^2^ values dropped below 0.3 at approximately 969,942 base pairs (Figure 3). Therefore, a window size of approximately 0.97 Mb was considered appropriate for identifying candidate genes in our population. This distance ensures that the SNP markers effectively capture causal variants’ effects, maintaining a sufficient level of LD.

The genomic association of SNPs detected by BLINK and FarmCPU and their nearby candidate genes are presented in Table 3.

Significant SNPs were identified using FarmCPU and BLINK models (Table 3). Notably, both models consistently detected significant SNPs on BTA 22 and 29: SNP ARS-BFGL-NGS-22211 (rs42598849) on chromosome 22 at position 43,274,058 showed a −log_10_*p* value of 6.1 with FarmCPU and −log_10_ *p* value of 7.6 with BLINK, and SNP ARS-BFGL-NGS-34508 (rs109387887) on BTA 29 at position 2,522,803 showed a −log_10_
*p* value of 7.2 with FarmCPU and −log_10_ *p* value of 8.0 with BLINK. The nearest gene to the SNP on BTA 22 was *DNASE1L3* (156,525 bp away from the SNP), and on BTA 29, the nearest gene to the significant SNP was *SLC36A4* (at a distance of 779,417 bp). Additionally, the BLINK model identified other significant SNPs: BovineHD4100011443 (rs41734577) on BTA 14 at position 29,888,019 (−log_10_ *p* = 6.3) was close to the *ARMC1* and *PDE7A* genes with a distance of 112,031 and 21,643 bp to the SNP, respectively. Hapmap48135-BTA-96568 (rs41591269) on BTA15 at position 5,170,124 (−log_10_ *p* = 6.2) was located near *MMP13* (at a distance of 500,747 bp to the SNP), and ARS-BFGL-NGS-39003 (rs110351063) on BTA 15 at position 66,127,887 (−log_10_ *p* = 7.6) was close to the *CD44* gene (distance of 390,425 bp to the SNP).

## 4. Discussion

In the present study, we performed a GWAS on variables derived from SCS in Slovenian Holstein Friesian cattle to investigate the association of genetic variants with these traits. This is the first study to utilize GWAS to analyze SCS traits in Slovenian Holstein Friesian population.

Our findings regarding SCS traits are consistent with those of previous studies. The overall mean and standard deviation (SD) of our defined traits align well with those reported in earlier studies. We calculated the mean SCS as 2.49 with an SD of 1.07 for the first lactation and a mean of 2.87 with an SD of 1.31 for the second lactation. These results are very close to those reported by Ilie et al. [20], who found mean SCS values of 2.81 ± 1.72 and 3.06 ± 1.89 for the first and second lactation, respectively. Nani et al. [13] reported a mean and SD of 5.10 ± 1.14 for the arithmetic mean of the top three SCS values (TOP3) in the first lactation in Argentinian Holstein cows, while our study found a lower mean and SD of 3.68 ± 1.32 for the same trait. Additionally, the mean SCSMAX values in our study were 4.52 and 4.99 for the first and second lactation, respectively. This range is within the means reported by other authors, who found SCSMAX values ranging from 3.9 to 5.83 [10,13]. The higher values in the study by Nani et al. [13] could result from different climatic conditions affecting SCS. Similarly, a higher mean and SD of 3.50 ± 2.38 were reported for SCS in Holstein cows in Brazil [33]. On the other hand, Costa et al. [34] reported values similar to our findings, with an average SCS of 2.91 ± 1.77 in Holstein cows from northern Italy. Lambertz et al. [35] concluded in their study on the impact of the temperature–humidity index (THI) on milk production traits and SCS that heat stress resulted in increased SCS.

For the association analysis in this study, we utilized FarmCPU and BLINK models. These models are designed to provide true genetic associations by integrating fixed- and random-effects models and effectively minimize false positives by controlling for population structure and multiple testing issues [28,29]. Several studies have compared the effectiveness of various statistical methods in detecting significant signals in traits with varying levels of heritability. For instance, Cebeci et al. [36] compared the performance of statistical methods used in GWAS on nine simulated quantitative traits with various degrees of heritability using genotypic data of domestic goats, and they suggested FarmCPU and BLINK as the best methods for controlling false positives and detecting true positive signals. In particular, the BLINK method was strongly recommended for GWAS analyses of low heritability traits, while both BLINK and FarmCPU were recommended for medium- and high-heritability traits. In addition, Kaler et al. [37] concluded that the FarmCPU model is highly effective for association mapping (AM) of complex traits in plants, as it controls both false positives and false negatives better than other models, including MLM, CMLM, ECMLM, MLMM, ANOVA, GLM, and SUPER.

In this study, we used three variables from SCS, including LM_SCS, MAXSCS, and TOP3, to test the association using FarmCPU and BLINK models. The lactation average SCS has been used for many years to analyze genetic parameters for SCS and in association studies, as this trait effectively reflects the variation in somatic cell counts over the lactation, providing valuable insights into the udder health [7,12,13,38,39]. However, Green et al. [14] suggested that the maximum value of SCS (MAXSCS) improves the prediction of clinical mastitis better than the lactation average SCS. In addition, Yamaguchi et al. [40] and Koeck et al. [10] estimated higher heritability of MAXSCS than for lactation average SCS. The TOP3 trait was first used by Nani et al. [13] in an association study for SCS in Argentinian dairy cattle to capture different aspects of mastitis.

For GWAS, we considered only the first two lactations. Yamaguchi et al. [40] highlighted the importance of considering different lactations as potentially distinct traits due to differences in the genetic architecture underlying traits like SCS across lactations. In particular, they found that genetic correlations between SCS traits in the first and second lactations were close to 1.0, suggesting a strong relationship between these traits across different lactations. We detected five significant SNPs associated with only the TOP3 trait in the first lactation on BTA 22, 29, 14, and 15. However, we did not detected significant signals associated with other traits, including SCSMAX and LM_SCS. None of the five significant SNPs identified in our study correspond to those reported in previous research [13,20,41,42].

The BTAs identified in this study have previously been reported to be associated with SCS or mastitis in other populations (Table 4). For example, Lund, M.S. et al. [43] confirmed the presence of a QTL affecting udder-health-related traits on BTA 14 in Nordic cattle breeds. Similarly, X. Wang et al. [44] detected ten significant SNPs associated with SCS on BTA 14 in the Chinese Holstein population; among these, two genes (*TRAPPC9* and *ARHGAP39*) were associated with these SNPs. In our study, we identified a novel SNP (BovineHD4100011443) at position 29,888,019 on BTA 14, which lies within a region spanning 7.86 Mb to 39.55 Mb, known to harbor QTLs potentially influencing both SCS and clinical mastitis [45]. Furthermore, the detection of SNPs on BTA 15 and 22 aligns with previous studies that have identified QTLs in these regions. For instance, a QTL on BTA 15 has been associated with clinical mastitis during the first lactation, and a QTL on BTA 22 has been detected for SCS in the Danish Holstein Cattle population [46]. Boichard et al. [47] detected a strong QTL on BTA 15 associated with SCS in French dairy cattle breeds. Additionally, a GWAS on Valdostana Red Pied cattle identified a significant QTL for SCS on BTA 15, specifically between 50.43 and 51.63 Mb [48]. Our SNP on BTA 15 (ARS-BFGL-NGS-39003), located at 66.13 Mb, was close to the range identified by this study. On BTA 22, a significant QTL for SCS was identified in the region spanning 52.95–53.26 Mb, which includes the *LTF* gene, known for its role in the immune response to bacterial infection in the mammary gland [49]. Our SNP on BTA 22 (ARS-BFGL-NGS-22211), located at 43.27 Mb, was approximately 9.7 to 10 Mb away from this QTL region. However, this same SNP, ARS-BFGL-NGS-22211 (rs42598849), has been previously reported for its association with body weight (BW) and body weight gain (BWG) [50].

Few studies have reported QTLs on BTA 29 for mastitis-related traits. For example, Schulman et al. [51] and Ashwell et al. [52] identified QTLs affecting SCS on several chromosomes, including BTA 29 in Finnish Ayrshire and Holstein cattle. However, Lund, M.S. et al. [43] could not confirm QTLs on BTA 29 affecting SCS or mastitis in Nordic dairy cattle breeds. Cai et al. [49] also noted the absence of reported QTLs for BTA 29 in the AnimalQTLdb. Our identified SNP, ARS-BFGL-NGS-34508 (rs109387887), on BTA 29 is located in a genomic region (2293, 9,642,553,781 bp) that had previously been reported as a QTL associated with body weight (BW) in cattle [53].

**Table 4 animals-14-02713-t004:** Previously reported QTL associated with significant SNPs.

SNP	BTA	Position (bp)	Previously Reported QTL	Trait	Reference
ARS-BFGL-NGS-22211 (rs42598849)	22	43,274,058	QTL near 52.95–53.26 Mb, associated with SCS and clinical mastitis	Somatic cell score, mastitis	[49]
ARS-BFGL-NGS-34508 (rs109387887)	29	2,522,803	QTL associated with body weight (BW) and body weight gain (BWG)	Body weight	[50]
BovineHD4100011443 (rs41734577)	14	29,888,019	QTL at 7.86 Mb–39.55 Mb region associated with SCS and mastitis	Somatic cell score, mastitis	[44]
Hapmap48135-BTA-96568 (rs41591269)	15	5,170,124	QTL between 50.43–51.63 Mb associated with SCS in Valdostana Red Pied cattle	Somatic cell score	[48]
ARS-BFGL-NGS-39003 (rs110351063)	15	66,127,887	QTL associated with fat percentage and fat yield traits in dairy cattle	Fat percentage, yield	[54]

The significant SNP on BTA 22, ARS-BFGL-NGS-22211 (rs42598849) was close to the gene *DNASE1L3* (distance from SNP: 156,525 bp), which encodes the enzyme deoxyribonuclease 1-like 3. This enzyme is a part of the DNase family, playing a crucial role in DNA fragmentation during processes such as apoptosis and in immune response, particularly in B-cell activation and antibody production [55,56]. In addition, this gene is essential for T cell-independent type II (TI–II) antigen response, which is vital for generating antibodies against specific pathogens [56]. *DNASE1L3* ensures controlled activation of inflammasomes, leading to regulated secretion of pro-inflammatory cytokines such as IL-1β and IL-18, which are important for the inflammatory response and signaling in immune cells [55]. Despite the significant involvement of *DNASE1L3* in immune regulation, which makes it a potential candidate in cattle immune response against mammary infections, there are a lack of studies investigating the role of this gene in cattle populations.

On BTA 29, the nearest gene to SNP ARS-BFGL-NGS-34508 (rs109387887) was *SLC36A4*, located 779,417 bp away from the SNP. *SLC36A4* is involved in the transport of small neutral amino acids across cell membrane, contributing to the cellular amino acid pool, which influences cellular metabolism and energy production [57]. Metabolite transporters like *SLC36A4* play a crucial role in immune response by facilitating the uptake of amino acids and key metabolites like kynurenine, which influence T-cell differentiation and activation [58]. In cattle, it has been shown that *SLC36A4* is more abundant in the mammary gland of cows with a genetic predisposition to high milk urea (HMUg) compared to those with low-milk-urea genetic predisposition (LMUg), suggesting an enhanced capacity for amino acid transport in HMUg cows, which can facilitate greater amino acid uptake for milk protein synthesis [59].

Another significant SNP, BovineHD4100011443, on BTA 14 is located near the Armadillo Repeat Containing gene 1 *(ARMC1)* (distance: 112,031 bp). The *ARMC* gene family is crucial for various cellular processes, primarily through its role in protein–protein interactions. These genes are involved in the regulation of the cell cycle, cell adhesion, and signal transduction [60]. *ARMC* family members are integral to maintaining a functional and responsive immune system. The *ARMC5* gene is involved in the regulation of the T-cell receptor (TCR) signaling pathway, which is crucial for T-cell development, activation, and differentiation [61]. In livestock, the *ARMC1* gene has been identified as a candidate gene associated with economically important traits in cattle, such as milk production, body weight, and protein production traits [62,63]. In addition, a few studies on mammary gland transcriptome and proteome in sheep and cows with clinical mastitis have detected *ARMC1* as a differentially expressed gene, indicating its potential involvement in the immune response to mammary gland infections [64,65,66]. Another gene, close to BovineHD4100011443 (BTA 14), was Phosphodiesterase 7A (*PDE7A)* (distance to SNP: 216431 bp). *PDE7A* is expressed in immune cells and plays a role in immune response. It modulates the activity of immune cells such as T cells and macrophages by affecting cAMP levels, which in turn influence cell activation, differentiation, and cytokine production [67]. Moreover, *PDE7A* is ubiquitously expressed in human proinflammatory and immune cells, suggesting its involvement in regulating inflammatory responses [68]. These findings could justify the role of *PDE7A* in cattle mastitis, which is an inflammatory response.

On BTA 15, we identified two significant SNPs. Hapmap48135-BTA-96568 (rs41591269) was close to the *MMP13* gene (distance from SNP: 500,747 bp). This gene is a member of the matrix metalloproteinases (MMPs) family of enzymes responsible for the breakdown of extracellular matrix (ECM) components. This enzyme plays a critical role in various physiological and pathological processes, including tissue remodeling, inflammation, and cancer progression [69]. It has been shown that the activation of the urokinase-type plasminogen activator system (uPA) and its effect on *MMP* levels, including *MMP13*, play a significant role in *S. aureus*-induced changes in bovine mammary fibroblasts (BMFBs) [70]. In addition, in a mouse model of *S. aureus* mastitis, increased expression of MMPs, such as *MMP2* and *MMP9*, contributed to tissue degradation and inflammation [71].

Another SNP at a different location on BTA 15 is ARS-BFGL-NGS-39003. Based on the animal QTL database, this SNP has been associated with fat percentage (FP) and fat yield (FY) traits [54]. Interestingly, this SNP was close to the gene *CD44* (distance from SNP: 390425), a well-known gene associated with inflammatory diseases. The *CD44* gene encodes a transmembrane glycoprotein involved in cell–cell interactions, cell adhesion, and migration. It plays a significant role in many cellular functions, including lymphocyte activation, recirculation, and tumor metastasis [72]. It has been shown that *CD44* plays an important role in recruiting blood polymorphonuclear neutrophil leukocytes (PMN) into the mammary gland in bovine mastitis and is significantly upregulated during mastitis [73]. In addition, *CD44* expression is significantly higher on neutrophils in both blood and milk during subclinical mastitis compared to clinical mastitis [74].

While our study identified significant SNPs associated with the TOP3 trait—a trait derived from SCS—in the first lactation on BTA 14, 15, 22, and 29, none of these SNPs corresponded to those reported in previous research [13,20,23,42]. There are several potential reasons for these discrepancies. First, the genetic background of the Slovenian Holstein Friesian cattle population may differ from those in other studies, leading to the variation in allele frequencies and the detection of different SNPs [75]. Additionally, the statistical models and methodologies used in GWAS can yield different results. Using FarmCPU and BLINK models effectively controls population structure and multiple testing [36]. Moreover, phenotypic definitions and measurements also play a crucial role. Variations in how SCS and mastitis are recorded and analyzed can lead to discrepancies in GWAS results [12,15,76]. Differences in sample size and statistical power between studies can impact the ability to detect significant associations, with smaller studies potentially missing SNPs that larger studies can identify [77,78].

In this study, we identified novel SNPs and candidate genes associated with SCS and mastitis resistance that have not been reported in previous research. These findings highlight unique genetic variation potentially contributing to mastitis resistance. To validate these results, further studies are necessary to examine the expression of these genes in cows with clinical and subclinical mastitis and in healthy cows. This additional research will help to understand the role of these novel genetic markers in mastitis resistance.

## 5. Conclusions

In this study, we conducted a GWAS on Slovenian Holstein Friesian cows to explore the genetic background of mastitis resistance. Among the three analyzed traits, LM_SCS, SCSMAX, and TOP3, we found associations only with the TOP3 trait in the first lactation. Our analysis identified several novel SNPs and candidate genes that have not been previously reported in other studies. These findings suggest genetic variation that could enhance mastitis resistance. The significant SNPs identified on chromosomes 14, 15, 22, and 29—along with their nearby candidate genes such as *DNASE1L3*, *SLC36A4*, *ARMC1*, *PDE7A*, *MMP13*, and *CD44*—which are associated with immune response and inflammatory processes present promising targets for further research. Future studies should focus on examining the expression levels of these genes in cows with clinical mastitis, subclinical mastitis, and healthy cows. This will help to validate our findings and to determine whether these novel genetic markers can be effectively utilized in breeding programs to improve mastitis resistance in dairy cattle. Integrating these markers into genomic selection strategies could enhance mastitis resistance, benefiting both herd health and milk production quality. Our study was conducted on a relatively small population of cows on a single farm. This limitation may affect the generalizability of our findings to other populations or breeds. Therefore, future studies should include a larger population of Slovenian Holstein cattle to validate our results and ensure broader applicability.

## Figures and Tables

**Figure 1 animals-14-02713-f001:**
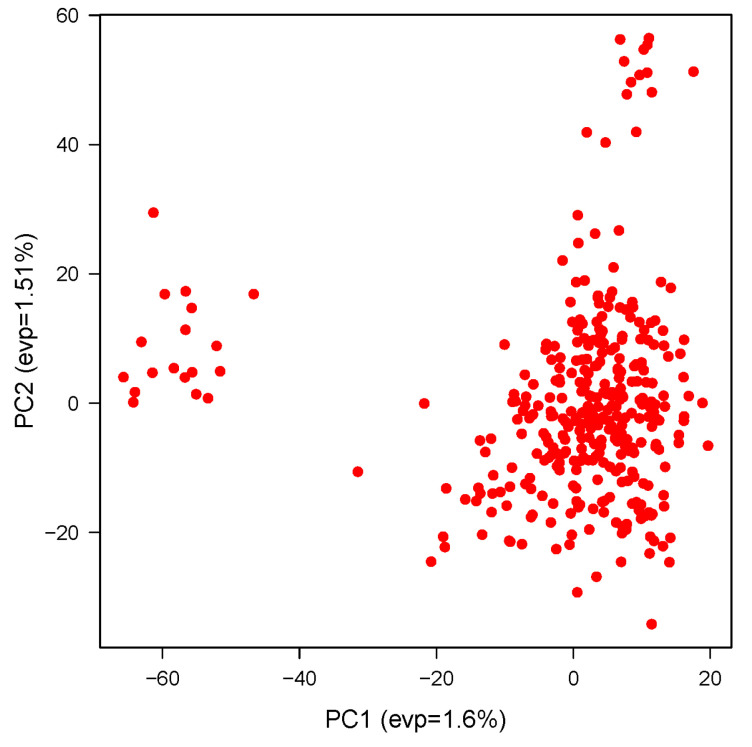
Principal component analysis (PCA) plot showing population structure in the analyzed herd of Slovenian Holstein Friesian Cows.

**Figure 2 animals-14-02713-f002:**
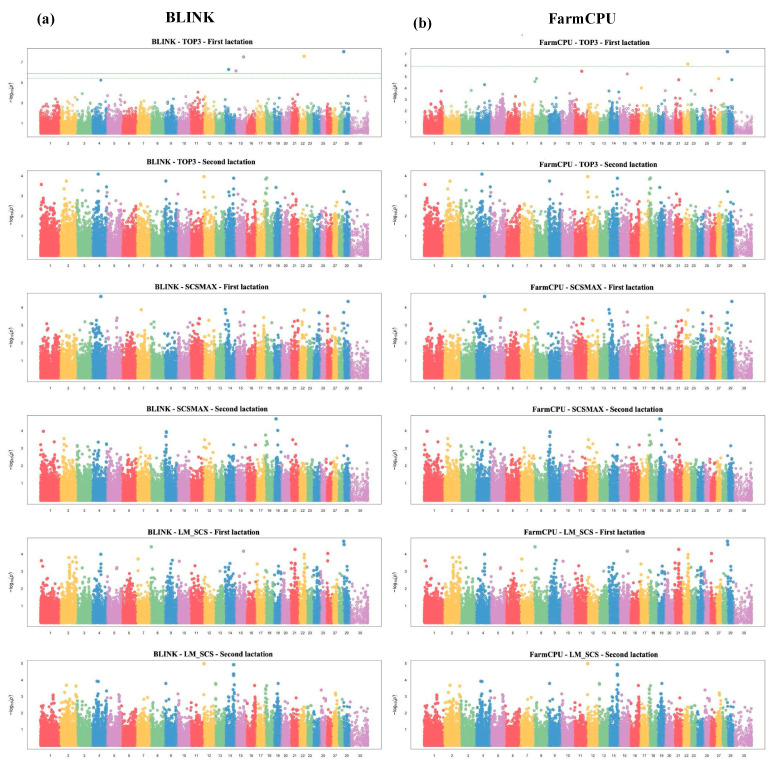
The Manhattan plot for the GWAS of SCS traits. Each chromosome (*x*-axis) is represented by a different color, and the plot is based on −log _10_ (*p*-value) from the GWAS against chromosome position. The green line indicates the genome-wide significant threshold at *p* = 1 × 10^−6^. SNPs identified by BLINK (**a**) and FarmCPU (**b**) models.

**Figure 3 animals-14-02713-f003:**
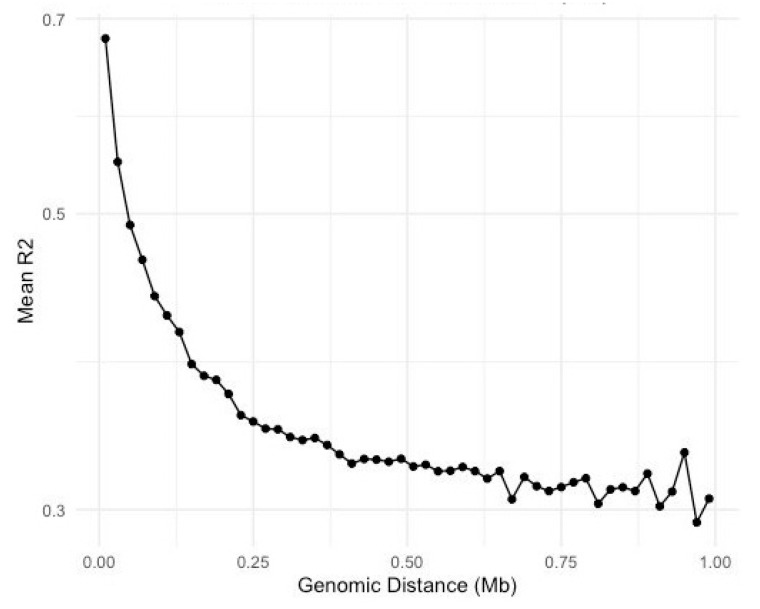
LD decay for distances between SNPs.

**Table 1 animals-14-02713-t001:** Summary statistics of SNP markers at *B. taurus* chromosomes ^a^.

Chr.	Size (Mb)	Number of SNP	SNP Density (SNP/Mb)	Mean Distance ± SD ^b^ (Kb)
NULL	-	83	-	-
1	158.53	2532	15.97	62.45 ± 60.32
2	136.23	2054	15.08	66.46 ± 68.81
3	121.01	2002	16.54	61.39 ± 70.7
4	120	1902	15.85	63.45 ± 55.88
5	120.09	1688	14.06	71.77 ± 73.77
6	117.81	1944	16.5	62.35 ± 81.38
7	110.68	1644	14.85	68.39 ± 76.54
8	113.32	1800	15.88	62.86 ± 55.08
9	105.45	1606	15.23	65.71 ± 67.11
10	103.31	1585	15.34	65.75 ± 102.79
11	106.98	1704	15.93	62.91 ± 58.9
12	87.22	1265	14.5	71.86 ± 117.62
13	83.47	1340	16.05	62.45 ± 56.94
14	82.4	1341	16.27	62.05 ± 59.99
15	85.01	1344	15.81	62.99 ± 67.81
16	81.01	1286	15.87	62.75 ± 77.98
17	73.17	1217	16.63	61.58 ± 65.78
18	65.82	1039	15.79	63.52 ± 77.78
19	63.45	1161	18.3	54.78 ± 54.44
20	71.97	1241	17.24	57.74 ± 55.56
21	69.86	1118	16	61.6 ± 58.43
22	60.77	966	15.9	63.44 ± 58.64
23	52.5	894	17.03	58.68 ± 62.61
24	62.32	948	15.21	65.58 ± 57.27
25	42.35	769	18.16	55.62 ± 44.6
26	51.99	811	15.6	62.91 ± 55.52
27	45.61	749	16.42	60.63 ± 72.18
28	45.94	734	15.98	62.84 ± 53.96
29	51.1	817	15.99	62.55 ± 74.74
X	139.01	323	2.32	459.62 ± 498.24
Overall	2628.38	39907	15.18	-

^a^ Based on ARS-UCD1.3. ^b^ Between adjacent SNPs.

**Table 2 animals-14-02713-t002:** Statistics for SCS traits in Slovenian Holstein Friesian cows.

Trait	Mean	Sd	Cv	Max	Min
Lm_scs_1_	2.49	1.07	0.43	5.8	0.48
Lm_scs_2_	2.87	1.31	0.53	7.63	0.1
Scsmax_1_	4.51	1.67	0.37	9.97	1.4
Scsmax_2_	4.99	1.9	0.42	9.9	0.53
Top3_1_	3.68	1.32	0.36	8.02	1.15
Top3_2_	4.17	1.57	0.43	8.53	0.53

Sd standard deviation; Cv, coefficient of variation; Max, maximum value; Min, minimum value; LM_SCS1 to 2, lactation mean SCS for first and second lactation; SCSMAX1 to 2, maximum SCS for first and second lactation; TOP3 1to2, mean of top 3 highest SCS values for first and second lactation.

**Table 3 animals-14-02713-t003:** Significant SNPs associated with the TOP3 trait in the first lactation.

SNP	Chr: Position	MAF	Alleles	Effect	−log_10_ (*p*-Value)	Model	Candidate Gene	Distance from SNP (bp)
BovineHD4100011443 (rs41734577)	14:29,888,019	0.26	A/G	−0.47	6.3	BLINK	*ARMC1, PDE7A*	112,031, 216,431
Hapmap48135-BTA-96568 (rs41591269)	15:5,170,124	0.23	A/G	0.49	6.2	BLINK	*MMP13*	500,747
ARS-BFGL-NGS-39003 (rs110351063)	15:66,127,887	0.45	G/A	0.46	7.5	BLINK	*CD44*	390,425
ARS-BFGL-NGS-22211 (rs42598849)	22:43,274,058	0.27	G/A	0.50	7.6 6.1	BLINK, FarmCPU	*DNASE1L3*	156,525
ARS-BFGL-NGS-34508 (rs109387887)	29:2,522,803	0.19	A/G	−0.59	8 7.2	BLINK, FarmCPU	*SLC36A4*	779,417

Note: MAF, minor allele frequency; Allele: the first allele is the nucleotide of the reference allele, and the second allele is the nucleotide of the alternate allele; Effect: the contribution of SNP to SCS; Model: the different models that successfully identified SNPs associated with the TOP3 trait in the first lactation; Candidate genes: the genes 1Mb upstream and downstream of the peak SNPs (reference genome, ARS-UCD1.3).

## Data Availability

Data are contained within the article.

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
