# Peer review of "Genome-Wide Association Study for Milk Somatic Cell Score in Holstein Friesian Cows in Slovenia"

_animals, 2024, doi:10.3390/ani14182713_

Round 1
Reviewer 1 Report
Comments and Suggestions for Authors
The manuscript entitled “Genome-wide association study for milk somatic cell score in 2 the Slovenian Holstein Friesian cows” by Ashja et al. is a standard GWAS study to identify the genome wide SNPs associated with somatic cell count-related traits, such as somatic cell score (SCS), lactation mean somatic cell score (LM_SCS), maximum SCS value (SCSMAX), and top 3 Mean value of SCS (TOP3), in Slovenian Holstein Friesian cows. The methods used in the study are appropriate. However, the results and discussion sections need improvement to represent the study to a greater level by mentioning many details. The present form of the manuscript undermines the quality of the work. Therefore, authors are suggested to improve the manuscript considering the following suggestions.
Introduction: The gaps in the existing literature on GWAS and mastitis are not clear in the introduction. This information can be provided to support the study. Were there not any similar studies on Slovenian Holstein Friesian cows?
Line 58: expand SCSSD
Materials and Methods;
1. How did authors identify the population structure? What is the relationship among all the animals in one farm? Show the relationship in pictorial form in the results.
Results:
1. The first paragraph of 3.2 appears to be the part of materials and methods. Hence, move this paragraph to subsection 2.1.
2. The study appeared to be performed on 3 somatic cell score traits, such as LMSCS, SCSMAX, and TOP3 in 2 lactations. However, the SNP association analyses results were shown only for the trait TOP3. What about the association results for other traits? Further, there is a typo mistake to mention 4 traits instead of 3 traits in the first sentence of the subsection 3.3. Please show the results of the remaining traits.
3. What are the heritability estimates in the studied dataset?
4. What is the correlation between the SNPs associated for a trait between two lactations?
5. As the study identified significant association between only 5 SNPs and TOP3 trait in the first lactation, it is better to show each genotype effect on the TOP3. Which allele of the SNP would be risk allele and which would be a resistant allele. Mention these details in a tabular form.
6. What are the previously reported QTLs present at the associated SNP loci. Mention these details in a tabular form.
Discussion:
1. The first paragraph is redundant. Similar information was mentioned in the introduction section. Hence, it can be omitted. If not, discuss this paragraph along with the heritability estimates calculated in the present study.
2. In the third paragraph, please explain the biological reasons for higher phenotypic parameters? Is there any chance of age effect? If so, the traits in the first and second lactations could be considered altogether different traits? As mentioned above, what is the correlation of SNP effects between both the lactations for a trait?
3. In paragraph 6, it was mentioned the inference of Yamaguchi et al.. However, what was the finding of this particular study? It is mentioned that “We detected no significant signals associated with 317 other traits, including SCSMAX and LM_SCS”. At what -log P value, this conclusion made?
4. The gene functions were well explained. However, these functions should be written in a refined manner towards their probable role either in the susceptibility or resistance to mastitis.
Comments on the Quality of English LanguageThere are some punctuation and organization problems in the text.
Author Response
Comment 1: The manuscript entitled “Genome-wide association study for milk somatic cell score in the Slovenian Holstein Friesian cows” by Ashja et al. is a standard GWAS study to identify the genome wide SNPs associated with somatic cell count-related traits, such as somatic cell score (SCS), lactation mean somatic cell score (LM_SCS), maximum SCS value (SCSMAX), and top 3 Mean value of SCS (TOP3), in Slovenian Holstein Friesian cows. The methods used in the study are appropriate. However, the results and discussion sections need improvement to represent the study to a greater level by mentioning many details. The present form of the manuscript undermines the quality of the work. Therefore, authors are suggested to improve the manuscript considering the following suggestions.
Introduction: The gaps in the existing literature on GWAS and mastitis are not clear in the introduction. This information can be provided to support the study. Were there not any similar studies on Slovenian Holstein Friesian cows?
Response 1: We have added sentences to clarify that, despite the widespread use of GWAS to identify genetic variants linked to mastitis and its indicator traits (such as SCS), no studies have explored the genetic architecture of SCS in Slovenian Holstein Friesian cows to the best of our knowledge. Additionally, we added the study on somatic cell count index in Slovenian Holstein Friesian cattle and mentioned that most of the previous GWAS studies in this population have focused on other (behavior) traits in two Slovenian cattle breeds.
Comment 2: Line 58: expand SCSSD
Response 2: We have expanded the "SCSSD" as standard deviation of SCS (SDSCS).
Comment 3: Materials and Methods;
- How did authors identify the population structure? What is the relationship among all the animals in one farm? Show the relationship in pictorial form in the results.
Response 3: Population structure was included in the model through the use of principal components (PCs) derived from principal component analysis (PCA). The PCs were included as covariates in the GWAS model to correct for population stratification. We added PCA plot to the results section (Figure 1).
Comment 4: Results:
- The first paragraph of 3.2 appears to be the part of materials and methods. Hence, move this paragraph to subsection 2.1.
Response 4: We moved the paragraph of 3.2. to 2.1.
Comment 5: The study appeared to be performed on 3 somatic cell score traits, such as LMSCS, SCSMAX, and TOP3 in 2 lactations. However, the SNP association analyses results were shown only for the trait TOP3. What about the association results for other traits? Further, there is a typo mistake to mention 4 traits instead of 3 traits in the first sentence of the subsection 3.3. Please show the results of the remaining traits.
Response 5: We corrected the typo, and expanded Fig.2 with association results for remaining traits.
Comment 6: What are the heritability estimates in the studied dataset?
Response 6: Heritability (30.6 %) was estimated using GAPIT3 based on mixed linear model using whole marker data. SNP-based heritability estimates only capture the genetic variance explained by the SNPs included in the dataset, potentially leading to an underestimation of true heritability, therefore we did not include heritability estimates in the manuscript.
Comment 7: What is the correlation between the SNPs associated for a trait between two lactations?
Response 7: We identified significant SNPs for the TOP3 trait in the first lactation, but none of these SNPs was consistently significant in the second lactation.
Comment 8: As the study identified significant association between only 5 SNPs and TOP3 trait in the first lactation, it is better to show each genotype effect on the TOP3. Which allele of the SNP would be risk allele and which would be a resistant allele. Mention these details in a tabular form.
Response 8: We have inserted a new figure with the results for all traits across both lactations (Figure 2). Additionally, we have provided a detailed table in the manuscript that outlines the effect of each significant SNP on the TOP3 trait (Table 3).
Comment 9: What are the previously reported QTLs present at the associated SNP loci. Mention these details in a tabular form.
Response 9: We have now added a table (Table 4) summarizing the previously reported QTLs present at the associated SNP loci.
Comment 10: Discussion: The first paragraph is redundant. Similar information was mentioned in the introduction section. Hence, it can be omitted. If not, discuss this paragraph along with the heritability estimates calculated in the present study.
Response 10: We deleted the first paragraph of the discussion.
Comment 11: In the third paragraph, please explain the biological reasons for higher phenotypic parameters? Is there any chance of age effect? If so, the traits in the first and second lactations could be considered altogether different traits? As mentioned above, what is the correlation of SNP effects between both the lactations for a trait?
Response 11: We have added a more detailed explanation regarding the higher phenotypic parameters observed. While we did not calculate the SNP effect correlation between the first and second lactations in our study, previous research has shown high genetic correlations between traits measured across lactations, often close to 1.0 (Yamaguchi et al., 2019).
Comment 12: In paragraph 6, it was mentioned the inference of Yamaguchi et al.. However, what was the finding of this particular study? It is mentioned that “We detected no significant signals associated with 317 other traits, including SCSMAX and LM_SCS”. At what -log P value, this conclusion made?
Response 12: The study by Yamaguchi et al. (2019) highlighted the importance of considering different lactations as potentially distinct traits due to differences in the genetic architecture underlying traits like SCS (Somatic Cell Score) across lactations. In particular, Yamaguchi et al. found that genetic correlations between SCS traits in the first and second lactation were close to 1.0, suggesting a strong relationship between these traits across different lactations.
Regarding the threshold for significance in our study, we applied a -logP threshold of 6.0 (p = 1 × 10^-6).
Comment 13: The gene functions were well explained. However, these functions should be written in a refined manner towards their probable role either in the susceptibility or resistance to mastitis.
Response 13: We have refined the explanation of gene functions to highlight their probable roles in either susceptibility or resistance to mastitis. In the revised manuscript, we have now linked the biological functions of genes to mechanisms relevant to immune response, inflammation, and tissue repair in the context of mastitis.

Reviewer 2 Report
Comments and Suggestions for Authors
I have reviewed the manuscript entitled "Genome-wide association study for milk somatic cell score in the Slovenian Holstein Friesian cows" from Animals with No-3183261. Below is a summary of the strengths and weaknesses, along with specific suggestions for improvement.
The study addresses an important issue in dairy farming—mastitis resistance—by identifying genetic markers associated with somatic cell score (SCS) in Slovenian Holstein Friesian cows. This is highly relevant to the field of animal breeding and genetics. Furthermore, the use of genome-wide association studies (GWAS) and robust statistical models (FarmCPU and BLINK) is appropriate and well-executed. The quality control measures for the genotypic data are thorough, ensuring reliability in the results. Finally, the study examines multiple SCS traits, providing a more nuanced understanding of the genetic factors associated with mastitis resistance. Overall, the results section is well-organized, with clear tables and figures that effectively communicate the findings.
However, for this article to be accepted and published, I personally have the following suggestions.
Title section: The phrase ‘in the Slovenian Holstein Friesian cows’ in the title should be changed to 'in the Holstein Friesian cows from Slovenia' because the cows are raised in Slovenia, rather than it being part of the breed name.
Introduction and Background section: Depth of Literature Review: While the introduction provides a good overview of the problem, it could benefit from a more detailed discussion of recent advances in GWAS for mastitis resistance. Highlighting the gaps in current knowledge that this study aims to fill would strengthen the rationale. Hypothesis: The introduction could more clearly state the specific hypotheses being tested. This would provide a stronger framework for the study.
Materials and Methods section: Sample Size and Generalizability: The study is based on a relatively small and homogeneous sample (350 cows from a single farm). Discussing the limitations this imposes on the generalizability of the findings to other populations or breeds would be important. Consider suggesting future studies that include larger, more diverse populations. Alternative SCC Traits: The choice of alternative SCC traits (LM_SCS, SCSMAX, TOP3) is well-justified. However, the rationale for choosing these specific traits over others could be expanded, potentially by referencing more studies that have successfully used these traits.
Results section: Significance Thresholds: The choice of the significance threshold (−log10P of 6.0) should be justified in the context of the study. It may also be helpful to discuss how this threshold was determined and whether it is consistent with other studies in the field. Interpretation of SNPs: While the study identifies several significant SNPs and candidate genes, the biological implications of these findings could be explored in more depth. Discussing the potential roles of these genes in mastitis resistance, based on existing literature, would add value.
Discussion section: Comparative Analysis: The discussion could benefit from a more detailed comparison of the findings with those of previous studies. Are the identified SNPs and genes consistent with findings in other breeds or populations? Practical Applications: While the potential use of these markers in breeding programs is mentioned, the discussion could provide more concrete examples of how these findings could be applied in practice. This would enhance the study's relevance to industry stakeholders.
Conclusion section: Summary of Findings: The conclusion could more clearly summarize the key findings of the study and their implications. A brief discussion of future research directions would also be beneficial.
Specific concerns:
Expand Literature Review: Incorporate more recent studies on GWAS in dairy cattle to provide a comprehensive background.
Clarify Hypotheses: Clearly state the hypotheses or research questions at the end of the introduction.
Discuss Limitations: Acknowledge the limitations of the study, particularly the sample size and the only focus on a single farm, and suggest ways to address these in future research.
Deepen Gene Function Discussion: Include more detailed discussion on the biological functions of the identified candidate genes and their potential roles in mastitis resistance.
Broaden Practical Implications: Elaborate on how the findings could be integrated into breeding programs, possibly including a discussion on the economic benefits of using genetic markers for mastitis resistance.
Finally, after adopting these suggestion, I think that these revisions would enhance the clarity, relevance, and impact of your paper.
Author Response
Comment 1: I have reviewed the manuscript entitled "Genome-wide association study for milk somatic cell score in the Slovenian Holstein Friesian cows" from Animals with No-3183261. Below is a summary of the strengths and weaknesses, along with specific suggestions for improvement.
The study addresses an important issue in dairy farming—mastitis resistance—by identifying genetic markers associated with somatic cell score (SCS) in Slovenian Holstein Friesian cows. This is highly relevant to the field of animal breeding and genetics. Furthermore, the use of genome-wide association studies (GWAS) and robust statistical models (FarmCPU and BLINK) is appropriate and well-executed. The quality control measures for the genotypic data are thorough, ensuring reliability in the results. Finally, the study examines multiple SCS traits, providing a more nuanced understanding of the genetic factors associated with mastitis resistance. Overall, the results section is well-organized, with clear tables and figures that effectively communicate the findings.
However, for this article to be accepted and published, I personally have the following suggestions.
Title section: The phrase ‘in the Slovenian Holstein Friesian cows’ in the title should be changed to 'in the Holstein Friesian cows from Slovenia' because the cows are raised in Slovenia, rather than it being part of the breed name.
Response 1: We changed the title as suggested.
Comment 2: Introduction and Background section: Depth of Literature Review: While the introduction provides a good overview of the problem, it could benefit from a more detailed discussion of recent advances in GWAS for mastitis resistance. Highlighting the gaps in current knowledge that this study aims to fill would strengthen the rationale. Hypothesis: The introduction could more clearly state the specific hypotheses being tested. This would provide a stronger framework for the study.
Response 2: We have added two sentences to clarify the gaps in current knowledge that the current study aims to address.
Comment 3: Materials and Methods section: Sample Size and Generalizability: The study is based on a relatively small and homogeneous sample (350 cows from a single farm). Discussing the limitations this imposes on the generalizability of the findings to other populations or breeds would be important. Consider suggesting future studies that include larger, more diverse populations.
Response 3: We acknowledge that the current study was conducted on a relatively small and homogeneous sample of 350 Holstein Friesian cows from a single farm, which may limit the generalizability of the findings to other populations or breeds. This sample size, while providing valuable insights into the genetic architecture of mastitis resistance in Slovenian Holstein Friesian cows, may not fully capture the genetic diversity present in larger or more diverse populations. To overcome these limitations, future studies should include a larger and more diverse sample of cows.
Comment 4: Alternative SCC Traits: The choice of alternative SCC traits (LM_SCS, SCSMAX, TOP3) is well-justified. However, the rationale for choosing these specific traits over others could be expanded, potentially by referencing more studies that have successfully used these traits.
Response 4: The selection of Lactation Mean Somatic Cell Score (LM_SCS), Maximum Somatic Cell Score (SCSMAX), and Top 3 Mean Somatic Cell Score (TOP3) as traits for this study is based on their ability to capture different aspects of mastitis susceptibility and udder health in dairy cows. These traits offer a more nuanced understanding of somatic cell count (SCC) dynamics than simply using the lactation average.
Comment 5: Results section: Significance Thresholds: The choice of the significance threshold (−log10P of 6.0) should be justified in the context of the study. It may also be helpful to discuss how this threshold was determined and whether it is consistent with other studies in the field.
Response 5: We added the sentence about the choice of −log10P = 6.0 in our study.
Comment 6: Interpretation of SNPs: While the study identifies several significant SNPs and candidate genes, the biological implications of these findings could be explored in more depth. Discussing the potential roles of these genes in mastitis resistance, based on existing literature, would add value.
Response 6: In the revised manuscript, we have now linked the biological functions of genes to mechanisms relevant to immune response, inflammation, and tissue repair in the context of mastitis.
Comment 7: Discussion section: Comparative Analysis: The discussion could benefit from a more detailed comparison of the findings with those of previous studies. Are the identified SNPs and genes consistent with findings in other breeds or populations?
Response 7: We have now added a table (Table 4) summarizing the previously reported QTLs present at the associated SNP loci.
Comment 8: Practical Applications: While the potential use of these markers in breeding programs is mentioned, the discussion could provide more concrete examples of how these findings could be applied in practice. This would enhance the study's relevance to industry stakeholders.
Response 8: We have added a sentence discussing future research opportunities and larger-scale studies that could further validate these markers, facilitating their widespread use across diverse populations. Integrating these markers into genomic selection strategies could enhance mastitis resistance, benefiting both herd health and milk production quality.
Comment 9: Conclusion section: Summary of Findings: The conclusion could more clearly summarize the key findings of the study and their implications. A brief discussion of future research directions would also be beneficial.
Response 9: We have revised the Conclusion section to more clearly summarize the key findings of the study and added a brief suggestion for future research directions.
Comment 10: Specific concerns:
Expand Literature Review: Incorporate more recent studies on GWAS in dairy cattle to provide a comprehensive background.
Response 10: We have incorporated more recent studies on GWAS in dairy cattle to provide a comprehensive background on mastitis resistance research.
Comment 11: Clarify Hypotheses: Clearly state the hypotheses or research questions at the end of the introduction.
Response 11: We have clearly stated the hypotheses and research questions at the end of the introduction to provide a stronger framework for the study.
Comment 12: Discuss Limitations: Acknowledge the limitations of the study, particularly the sample size and the only focus on a single farm, and suggest ways to address these in future research.
Response 12: We acknowledged the limitations of our study, particularly the small sample size and the fact that the study was conducted on a single farm. We also suggested future research which could address these limitations, including larger, more diverse populations.
Comment 13: Deepen Gene Function Discussion: Include more detailed discussion on the biological functions of the identified candidate genes and their potential roles in mastitis resistance.
Response 13: We expanded the discussion of gene functions, explaining in more detail their biological roles and their potential influence on mastitis susceptibility and resistance.
Comment 14: Broaden Practical Implications: Elaborate on how the findings could be integrated into breeding programs, possibly including a discussion on the economic benefits of using genetic markers for mastitis resistance.
Response 14: We elaborated on the practical applications of the findings, specifically how they could be integrated into breeding programs. We also touched upon the potential economic benefits of using genetic markers to improve mastitis resistance, which would enhance herd health and milk production quality.

Round 2
Reviewer 1 Report
Comments and Suggestions for Authors
The manuscript entitled "Genome-wide association study for milk somatic cell score in Holstein Friesian cows in Slovenia" by Ashja et al. is significantly detailed after the revision, so that the manuscript is useful to many readers. Authors addressed most of the comments appropriately. The revised manuscript can be accepted for the publication.
Reviewer 2 Report
Comments and Suggestions for Authors
This revision meet my concerns so I would like to recommendate it for publiccation.